# Charge carrier localised in zero-dimensional (CH$_3$NH$_3$)$_3$Bi$_2$I$_9$ clusters

Chengsheng Ni[1,2], Gordon Hedley [3], Julia Payne[1], Vladimir Svrcek[4], Calum McDonald [5], Lethy Krishnan Jagadamma[3], Paul Edwards [6], Robert Martin [6], Gunisha Jain[5], Darragh Carolan [5], Davide Mariotti[5], Paul Maguire [5], Ifor Samuel [3] & John Irvine[1,7]

A metal-organic hybrid perovskite (CH$_3$NH$_3$PbI$_3$) with three-dimensional framework of metal-halide octahedra has been reported as a low-cost, solution-processable absorber for a thin-film solar cell with a power-conversion efficiency over 20%. Low-dimensional layered perovskites with metal halide slabs separated by the insulating organic layers are reported to show higher stability, but the efficiencies of the solar cells are limited by the confinement of excitons. In order to explore the confinement and transport of excitons in zero-dimensional metal–organic hybrid materials, a highly orientated film of (CH$_3$NH$_3$)$_3$Bi$_2$I$_9$ with nanometre-sized core clusters of Bi$_2$I$_9$$^{3-}$ surrounded by insulating CH$_3$NH$_3$$^+$ was prepared via solution processing. The (CH$_3$NH$_3$)$_3$Bi$_2$I$_9$ film shows highly anisotropic photoluminescence emission and excitation due to the large proportion of localised excitons coupled with delocalised excitons from intercluster energy transfer. The abrupt increase in photoluminescence quantum yield at excitation energy above twice band gap could indicate a quantum cutting due to the low dimensionality.

[1] School of Chemistry, University of St Andrews, Scotland KY16 9ST, UK. [2] College of Resources and Environment, Southwest University, Beibei, Chongqing 400716, China. [3] School of Physics and Astronomy, University of St Andrews, Scotland KY16 9ST, UK. [4] Research Center for Photovoltaics, National Institute of Advanced Industrial Science and Technology (AIST), Tsukuba 305-8568, Japan. [5] Nanotechnology and Integrated Bioengineering Centre, Ulster University, Northern Ireland BT37 0QB, UK. [6] Department of Physics, SUPA, University of Strathclyde, John Anderson Building, 107 Rottenrow, Glasgow, Scotland G4 0NG, UK. [7] Key Lab of Design and Assembly of Functional Nanostructure, Fujian Institute of Research on the Structure of Matter, Chinese Academy of Sciences, Fuzhou, Fujian 350002, China. Correspondence and requests for materials should be addressed to J.I. (email: jtsi@st-andrews.ac.uk)

Metal–organic hybrid perovskites share the formula of $ABX_3$ (A = organic group, B = metallic ion, X = anions) with a three-dimensional (3D) framework of corner-sharing $(BX_6)^{4-}$ octahedron, which is also the reason for high carrier mobility. The importance of the 3D framework to the transport of charge carriers can be evidenced by performance dependence on crystal orientation in layered two-dimensional (2D) perovskites[1]. For example, in the photovoltaic context, films of 2D Ruddlesden–Popper phases, with randomly distributed crystallites, have shown promising stability, but poor efficiency at only 4.73%[2] in contrast to the efficiency of around 20% in 3D perovskites[3], which could be attributed to the inhibition of out-of-plane transport due to the insulating organic groups[1, 2, 4, 5]. On the contrary, with careful control of crystallinity and anisotropy, a 2D perovskite can also show high performance close to 3D perovskites[1]. As a collateral effect of this confinement of charge carriers in the atom-scale slab that increases the exciton recombination, the low-dimensional layered perovskites with layered corner-sharing $(BX_6)^{4-}$ separated by organic cations have been reported to show structural relaxation and high luminescence when the thickness is decreased to few unit cells[6]. The dimensionality of a 2D perovskite can be reduced further to zero-dimensional clusters, noted as $(B_nX_m)^{j-}$ surrounded by $A^{k+}$ (m, n, j, k are integers), by tuning the crystallography and valence of cations and anions[7]. For example, organic–inorganic hybrid bismuth halide can generate a single $(BiX_6)^{3-}$ octahedron[8], face-sharing $(Bi_2I_9)^{3-}$ or edge-sharing $(Bi_2I_{10})^{4-}$ bioctahedra[9, 10], which have a size of ~1 nm separated by organic groups.

Nanometre-sized quantum dots (QDs) can display optical, electronic, and structural properties that often are not present in either isolated molecules or macroscopic solids[11]. For example, QDs of Si and PbS can efficiently generate multiple excitons from one high-energy absorbed photon following a mechanism known as carrier multiplication (CM) owing to their discrete energy levels and enhanced Coulomb interactions[12–14]. This is one of the suggested phenomena that may allow the efficiency of a single-junction solar cell to exceed the Shockley–Queisser limit[15]. The production of additional electron–hole pairs is expected to be greatly enhanced in QDs because their limited electron density of states greatly reduces the electron–phonon interactions[12]; however, for a solar cell using PbS QDs as an absorber, the large number of higher-lying energy levels due to surface defects were reported to decrease CM[16]. This could be alleviated by surface passivation with ligands[17], alternatively this might also be achieved by protecting with the organic cation in zero-dimensional hybrid organic–inorganic halides. Moreover, the intrinsic junction between the negative charge on the low-dimensional $(B_nX_m)^{j-}$ cluster and opposite charge on the surrounding organic $A^{k+}$ could cause an internal electric field that controls the broadband emission that could be tuned via the electronegativity of the core cluster and surrounding organic groups[18].

The coupling between QDs and perovskite is of interest since the heterocrystals of QDs embedded in perovskites exhibit remarkable optoelectronic properties that are traceable to their atom-scale crystalline coherence that involves the efficient transport of excitons between them[19]. A metal–organic material with the formula $(CH_3NH_3)_3Bi_2I_9$ (MABI) that contains quantum confined clusters/dots of $(Bi_2I_9)^{3-}$ has been reported as an effective absorbing material in a solar cell, but the performance was low, 0.1%[20, 21]. With a small size (around 1 nm) of $(Bi_2I_9)^{3-}$, MABI showed a separate excitonic peak in the absorption spectra due to the confinement of charge carriers, and the binding energy of charge carriers has been estimated to be over 0.3 eV[22]. An analogous explanation on the even poorer performance compared to the 2D layered perovskite could be the

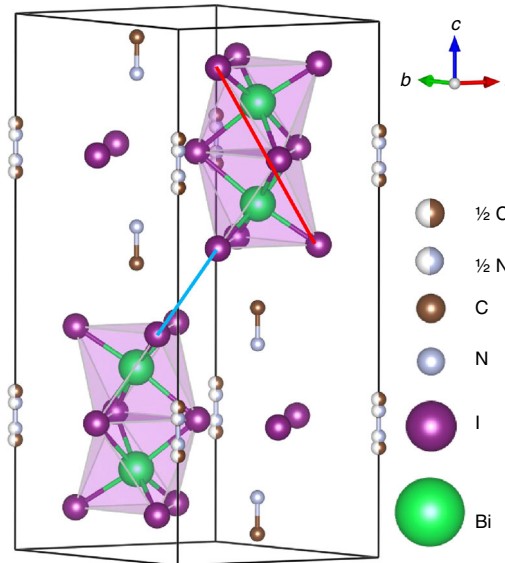

**Fig. 1** Crystal structure of $(CH_3NH_3)_3Bi_2I_9$ from single-crystal diffraction The hydrogen atoms are omitted. The *thick light-blue line* (0.43 nm) shows the shortest distance between two $Bi_2I_9$ bioctohedra and the *thick red line* (0.85 nm) shows the distance between two furthest iodine atoms. ½ C and ½ N represent the half-occupied carbon and nitrogen sites, respectively

enhanced confinement of charge carriers, but this explanation has not been elucidated because the performance of a solar cell depends on multiple parameters, such as the band energy alignment of different components and the microstructure of the layers. In a more recent study, MABI shows a double-peak emission including one close to band gap coupled with one in the blue light region and is superior in stability over lead-based perovskite in ambient air[23, 24]. Interestingly, long-lived excitons on the scale of nanoseconds have also been observed, which contrasts with the low performance of devices based on MABI[23, 24]. In this study, we synthesised a highly orientated MABI film to explore these issues. We studied the photoluminescence (PL) anisotropy and lifetime of excitons on these low-dimensional hybrid materials in order to understand the charge-carrier transport property. The PL at high excitation energy above twice the band gap ($E_g$) seems to show an enhancement in quantum yield that may be explained in terms of CM[25], which could open up a horizon of low dimensionally bonded hybrid materials for third-generation photovoltaics.

## Results

**Structure analysis of MABI crystal and orientated film.** Figure 1 shows the structure of MABI determined by single-crystal diffraction measured at room temperature[24]. The materials show hexagonal symmetry ($P\ 6_3/m\ m\ c$) and the face-shared $BiI_6$ bioctahedra or $(Bi_2I_9)^{3-}$ clusters are separated by $CH_3NH_3^+$ groups. The unit cell is $a = 8.5821(3)$ Å and $c = 21.7678(8)$ Å, leading to a volume of 1388.46(11) Å$^3$ and a density of 3.8 g cm$^{-3}$. Considering the size of an iodine anion, the $(Bi_2I_9)^{3-}$ clusters are *ca.* 1.11 nm in size and are separated from each other by *ca.* 0.16 nm. The density of the $(Bi_2I_9)^{3-}$ clusters is $1.69 \times 10^{27}$ cm$^{-3}$, which is much higher than the likely concentration of nanoparticles ($10^{17–20}$ cm$^{-3}$) suspended in solvent[25]. The thermal stability of MABI as a function of temperature was studied with differential scanning calorimetry, and no phase transformation was observed between −50 and 140 °C (Supplementary Fig. 1). This differs from the perovskite structure that changes from orthorhombic at room temperature to hexagonal or cubic

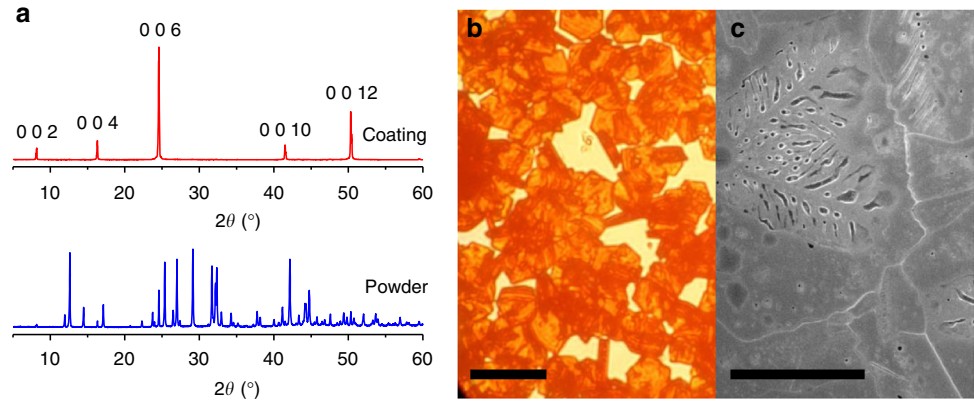

**Fig. 2** Structure characterisation of the orientated $(CH_3NH_3)_3Bi_2I_9$ coating on quartz. **a** X-ray diffraction results from the coating and powdered samples. **b** Optical microscope and **c** scanning electron micrograph of the $(CH_3NH_3)_3Bi_2I_9$ coating processed by depositing and drying of the solution of $BiI_3$ and $CH_3NH_3I$ in N,N-Dimethylmethanamide. The *scale bars* represent 25 μm

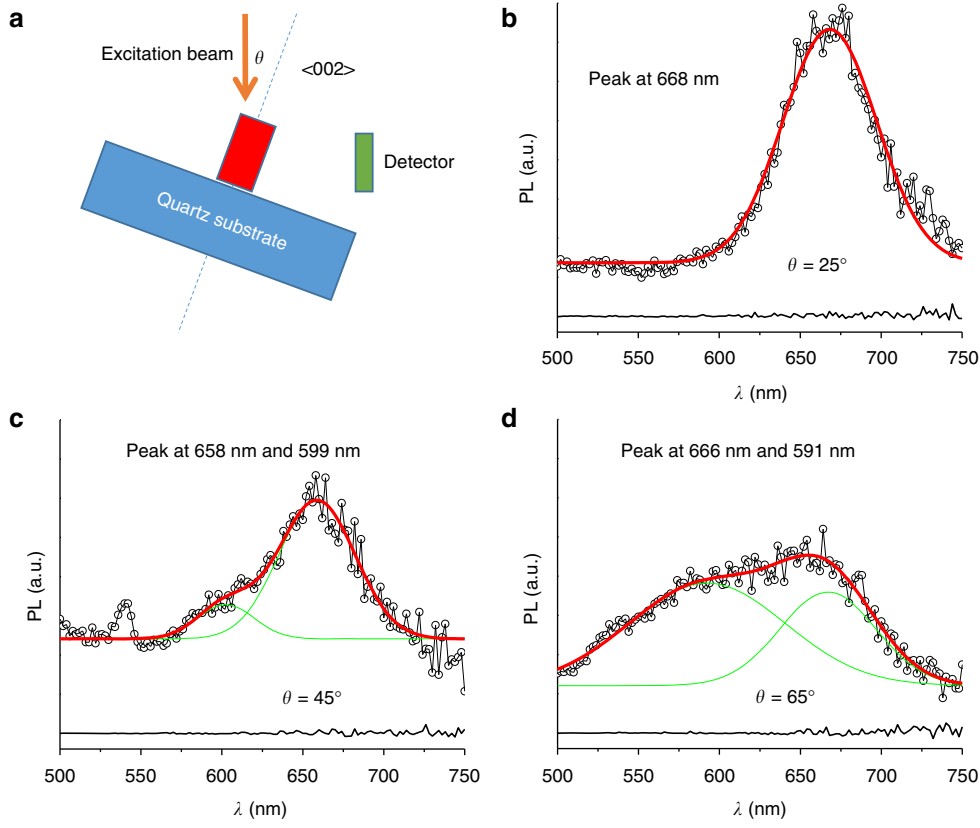

**Fig. 3** Photoluminescence (PL) anisotropy of $(CH_3NH_3)_3Bi_2I_9$ coating. **a** Experimental set-up and coating orientation. The detector and incident angle are positioned at right angles and the substrate is rotated to control the angle. The sample is positioned vertically and the wavelength of incident beam is 350 nm. **b**–**d** show the respective PL spectra at 25°, 45° and 65°. The *black lines* at the *bottom* of **b**–**d** are the PL measurement on the bare fused silica substrate at the respective angles

structure at higher temperature[26]. The stable structure of MABI could be beneficial in the stability and lifetime of the device. In previous studies involving Si nanocrystals with quantum cutting phenomena, the spatial distribution showed that 50% of the nearest-neighbour distances are below 1 nm[12]. The $CH_3NH_3^+$ group surrounding the $(Bi_2I_9)^{3-}$ clusters at the atomic level behaves like the atomic ligands that separates and passivates PbS QDs[17].

Figure 2 shows the results of X-ray diffraction (XRD), optical microscopy and scanning electron microscopy (SEM) obtained from the coating. The film is mostly composed of large grains

with good crystallinity. The XRD pattern of a large area film shows only peaks in the highly orientated structure in the <002> direction, while the XRD of the powder by crushing the film showed more peaks from the other crystal planes. The refined unit cell parameters, $a = 8.5792(1)$ Å, $c = 21.7588(4)$ and $V = 1386.95(5)$ Å$^3$, are in good agreement with those previously reported for MABI[24]. The X-ray photoemission spectroscopy (XPS) and Raman spectroscopy on the prepared film (see Supplementary Figs. 2 and 3 and Supplementary Table 1) indicate that the surface of the film is not degraded in ambient air. The ultraviolet-visible spectroscopy (Supplementary Fig. 4) shows a

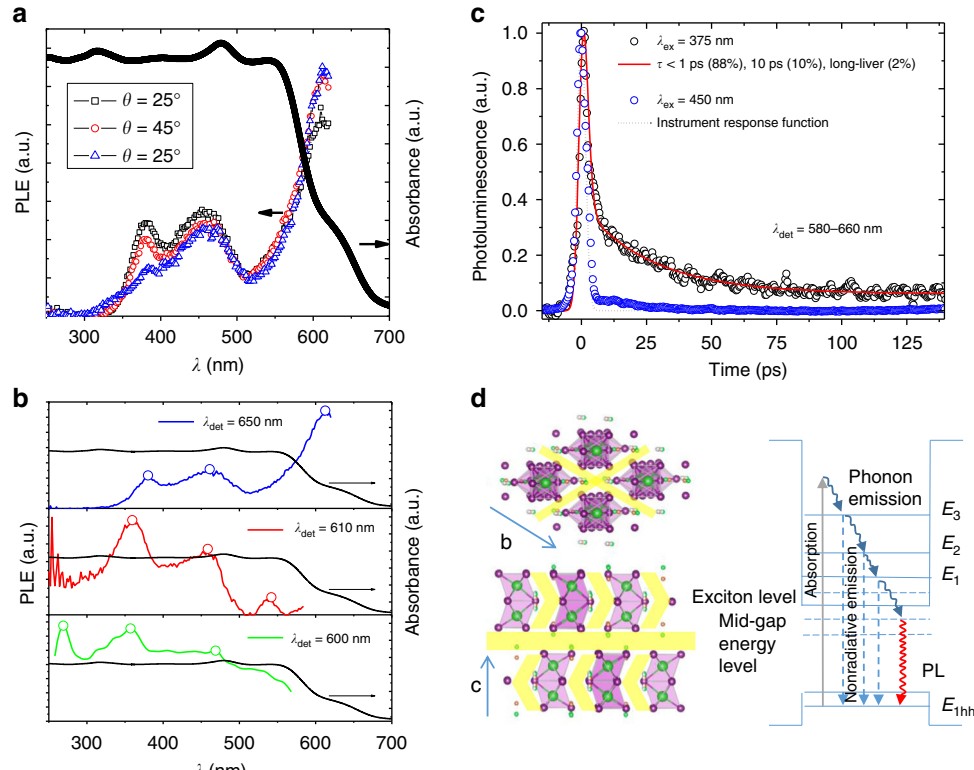

**Fig. 4** Excitation property of the charge carriers. **a** Ultraviolet–visible absorption derived from diffuse reflectance spectra of $(CH_3NH_3)_3Bi_2I_9$ and the angular dependence of photoluminescence excitation (PLE) of 650 nm emission ($\lambda_{det} = 650$ nm) at 25°, 45° and 65°. **b** PLE spectra for emission at $\lambda_{det} = 650$, 610 and 600 nm while keeping incident angle, $\theta$, at 45°. The *open circles* show the major PLE peaks. **c** Time-resolved photoluminescence of the coating placed at $\theta = 0°$ under a excitation beam of 375 nm ($\lambda_{ex} = 375$ nm) and 450 nm ($\lambda_{ex} = 450$ nm). **d** Schematics of quantum-well structure and multiphonon relaxation. The mid-gap energy level is from the organic termination. $E_{hh}$ represents the energy level of heavy hole state, while $E_1$, $E_2$ and $E_3$ show the discrete energy levels in the conduction band

band gap of 2.19 eV with a direct transition due to the excitonic nature of the absorption peak, similar to the value reported by Kawai et al. on MABI single crystal[22]. Although there was a report on the indirect transition of band gap for MABI based on the density functional theory calculations that imply a split of conduction band[21, 23], it should be noted that the absorption spectra of MABI is dependent on the orientation of crystals[24], which is a strong indication of charge carrier localisation. The absorption spectrum of a MABI film shows an excitonic peak at 2.44 eV, which is slightly different to that of a similar $(C_6H_{11}NH_3)_3Bi_2I_9$ (2.47 eV[22, 27]) containing $(Bi_2I_9)^{3-}$ clusters but with a larger organic group. Since this peak is usually assigned to the $(Bi_2I_9)^{3-}$ group[22], the organic group does not contribute to the absorption at this energy level. The lowest exciton state arises from excitations between the valence band, which consists of a mixture of Bi(6s) and I(5p) states, and the conduction band, which derives primarily from Bi(6p) states, and is confined zero-dimensionally in the bioctahedra $Bi_2I_9^{3-}$[27]. Substitution of Cs or formanidinium on methylamonium (MA) site in MAPbI$_3$ has been shown to cause significant shift (larger than 100 meV) of the band edge[20, 28], contrasting with MABI where the band gap is unchanged between Cs and MA[20].

**PL anisotropy of MABI coating**. The fundamental property of a crystalline material is the periodic spatial arrangement of atoms and clusters. The MABI crystals containing periodically aligned $Bi_2I_9^{3-}$ clusters could be viewed as QDs with organic groups as separators, resembling a core–shell structure due to the nanometre size of the photoactive $Bi_2I_9^{3-}$ clusters. Since the MABI crystals are aligned in the <002> direction on the quartz

substrate and orientated randomly in the perpendicular directions, there would be PL anisotropy in the <002> direction. PL anisotropy is investigated by varying the angle of the coating with the incident beam while fixing the position of the detector. Distinct PL spectra are obtained as shown in Fig. 3: at 25°, only one peak is shown, while at 45° and 65°, the spectra split into two peaks, and the relative height of the peak at the blue side tends to increase with the incident angle, $\theta$. On the other hand, the cathodoluminescence (CL) of MABI coating in Supplementary Fig. 5 does not show luminescence anisotropy but a continuous emission from 490 to 750 nm. One of the fundamental differences between CL and PL is that, while a photon in PL generates only one electron–hole pair, one 5-keV electron in CL, for example, can generate hundreds of electron–hole pairs in the excitation volume, which is usually several microns in diameter[29]. Specifically, the penetration depth of the electron beam of 5 keV would be around 200 nm based on the density of 3.8 g cm$^{-3}$. This difference makes the actual incident angle of a high-energy electron beam in CL very difficult to control because of the re-absorption of photons.

**Charge-carrier localisation and intercluster transport in $(CH_3NH_3)_3Bi_2I_9$.** The PL excitation (PLE) technique allows one to measure all the allowed intersub-band absorption transitions, and therefore to make a much more complete characterisation of low-dimensional system. The PLE spectra (Fig. 4a) for emission detection at 650 nm ($\lambda_{det} = 650$ nm) can be fitted by four peaks (Supplementary Fig. 6) centred, respectively, at 385, 464, 557 and 615 nm, showing a discrete absorption. The change of incident angle has a pronounced effect on the relative height of these

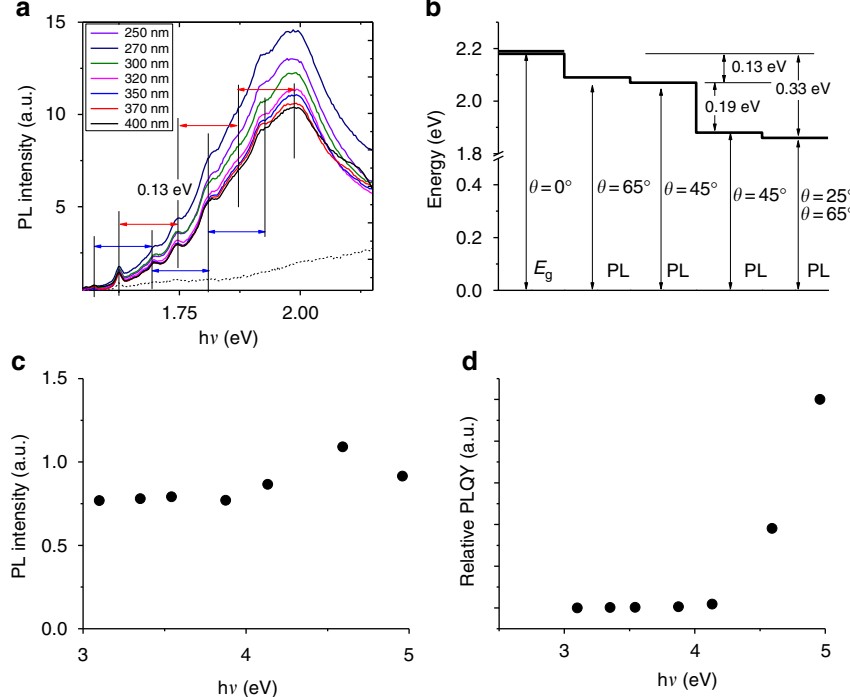

**Fig. 5** Photoluminescence (PL) at difference excitation energies. **a** PL spectra of the powdered (CH₃NH₃)₃Bi₂I₉ under different excitation energies. The *blue* and *red arrows* show the Stokes shift of 0.13 eV. The *dashed line* at the *bottom* is the PL measured on the substrate excited at 400 nm. **b** Energy diagram for the band gap of orientated film and the peak energy of PL at different angles. **c** Integration of PL intensities at different excitation energies. **d** Relative PL quantum yield (PLQY) for the (CH₃NH₃)₃Bi₂I₉ powder and coating on quartz, respectively

peaks, corresponding to the absorption variation with the crystal directions[24]. As anisotropic crystals have a light absorption coefficient depending on the direction of the light wave and its polarisation, the variations of peak heights with incident angle confirm the optical anisotropy of MABI. More importantly, the PLE spectra for different $\lambda_{det}$ (Fig. 4b) show distinct features, an indication of localised excitons that are not free to move into all parts of the QDs. As the $\lambda_{det}$ shifts to the blue side from 650 nm, the bleaching of low-energy absorption peak is probably associated with the well that is heavily electron-occupied[30] (as in the charge on Bi₂I₉³⁻) or the increasing emission overlap of trapped states.

The lifetime measurements on the excitons when the sample is placed perpendicular to the excitation beam ($\theta = 0°$; Fig. 4c) were performed using two excitation wavelengths ($\lambda_{ex} = 375$ or 450 nm) close to the PLE peaks, while detecting emission, $\lambda_{det}$, is between 580 and 660 nm. When excited at 375 nm the samples show very fast (life time of excitons, $\tau < 1$ ps), slower ($\tau = 10$ ps) and very slow ($\tau > 2$ ns) PL decays, while when excited with 450 nm light it is just very fast ($\tau < 1$ ps) and very slow ($\tau > 12$ ns). The picosecond excitons under 450 or 375 nm excitation could perhaps be inside the cluster of Bi₂I₉³⁻ and the distinct lifetime profile indicates that the emissions at these two excitations are from trapped states of different sites. The early-time emission (Supplementary Fig. 7) with both excitation wavelengths is centred on ~600 nm, while the long time 20-ps emission is centred on ~725 nm. Because the PL from the fluorometer (steady state) is an integral of what we see with the streak camera and only one peak at ~700 nm is shown at low incident angle, the long-lived excitons will contribute mostly to the PL spectra. The nanosecond lifetime of a small portion of excitons indicates perhaps that some of them tunnel through the gap of CH₃NH₃⁺ and become delocalised because of the interdot coupling[24].

For a QD, the cooling of carriers would require multiphonon processes when the quantised levels are separated in energy by more than the fundamental phonon energy, as shown in Fig. 4d. Excitons might travel between dots through multistep, phonon-assisted tunnelling, or homo-Förster resonance energy transfer that increases the lifetime of the acceptors. The tunnelling of excitons is very sensitive to the interdot distance. Showing the same type of absorptions, CdSe nanocrystallites in a closely packed superlattice show a red-shifted PL in a dilute colloidal system due to the interdot coupling[11, 31], but, in an anisotropic crystalline MABI, we demonstrated that the incident angle is an additional parameter for the emission spectra.

**PL quantum yield and quantum cutting**. Because of the high degree of orientation of the MABI film, the PL is anisotropic, so overall PL of the coating near the band gap is measured on a powder sample under different excitation beam, ranging from 400 to 250 nm, as shown in Fig. 5a. A set of parasitic peaks at the low-energy side of the main peak is an indication of cascade excitations, suggesting that the re-absorption and re-emission of photons among the micron-scale crystals is possible since the absorption edge from the diffuse reflectance spectroscopy of a powdered sample is as low as 1.77 eV (700 nm). Because of the low dimension of Bi₂I₉³⁻ (*ca.* 1 nm, close to that of a molecule), the Stokes shift causes lower-energy emission when an emitted photon from one crystal is reabsorbed by a neighbouring one. Because of the anisotropic nature of the Bi₂I₉³⁻ cluster, two main peaks rather than one are observed according to the anisotropic PL of a coating, and a Stokes shift energy of 0.13 eV can be obtained from the PL spectra. As the band gap of the highly orientated film ($\theta = 0°$) is determined to be 2.19 eV, while the peak energy of PL singlet at $\theta = 25°$ is 1.86 eV, giving a binding energy in this situation ~0.33 eV at low angles.

Kawai et al. studied the absorption of MABI single crystals of different thicknesses in the temperature range between 78 and 301 K, and the excitonic nature of the absorption peak at 2.43 eV was proven via the fitting of Urbach tail at the low-energy side of the peak and the binding energy was estimated to be higher than 0.3 eV[22]. The Stokes shift of the powdered sample is lowered to 0.13 eV because of the higher-energy level at higher angles, as illustrated in Fig. 5b. A relative PL quantum yield (PLQY)[25] under each excitation energy as shown in Fig. 5d was obtained by normalising the integration of PL spectra (Fig. 5c) by the pump intensity (Supplementary Fig. 8) and absorption. An increase in the PLQY was observed when the excitation energy is above 4.13 eV. The nearly linear increase in the quantum efficiency, rather than an abrupt jump, after the photon energy is higher than twice the band gap, $2E_g$, implies that the possible CM process happens possibly via the impact ionisation mechanism as simulated by Allan et al.[32] In MABI, a second $(Bi_2I_9)^{3-}$ may be excited by a neighbouring $(Bi_2I_9)^{3-}$ cluster containing a hot electron–hole pair, since the 0.11-nm separating distance controlled by the size of $CH_3NH_3^+$ groups is sufficiently small to facilitate the energy transfer. MABI contains a small number of delocalised excitons with lifetime on the scale of nanoseconds, but such systems can still be expected to produce CM[33]. The difference in the lifetime of excitons at $\lambda_{ex} = 450$ and 375 nm showed limited impact on either the PL intensity or PLQY. The Tauc plot (Supplementary Fig. 9) of bulk $CH_3NH_3I$ was presented to estimate the excitation energy of the organic separator, showing a large band gap of 4.83 eV, but MABI still loses its low-dimensionality in CL with excitation energy as high as 5 keV.

## Discussion

A low-dimensional hybrid material, MABI, showing a band gap of 2.19 eV and absorbing visible light has been obtained via solution processing. The film resulting from drying the solution showed a high degree of preferred orientation in the <002> direction. The close coordination between the negatively charged core cluster, $(Bi_2I_9)^{3-}$, surrounded by positive $CH_3NH_3^+$ seems to facilitate the localisation of excitons. The resultant anisotropic electron density would give rise to the Stokes shift in the PL. A small portion of long-lived excitons arises from the coupling and energy transfer between the clusters. The quantum yield from the QD core clusters separated by insulating organic cations is seen to increase when the excitation is above $2E_g$, indicating quantum cutting.

## Methods

**Materials**. $Bi_2O_2(CO_3)$ (2 g, Alfa Aesar 99.9%) was dissolved in hydroiodic acid (HI) (10 ml, Alfa Aesar) using ethanol as solvent (500 ml) and then the product, $BiI_3$, collected after the solution was evaporated at 80 °C. Methylammonium iodide was collected by reacting methylammonia (24 ml) solution in hydriodic acid solution (10 ml) in ethanol (100 ml). The solvent was removed at 50 °C using a rotary evaporator. The resulting white solid was washed in diethylether and dried in a vacuum oven overnight.

**Characterisation of coating and powdered sample**. $BiI_3$ and $CH_3NH_3I$ (1:2 in mole ratio) were dissolved in dimethyl formamide (DMF) to form a clear solution (20 wt.% of salts) when heated at 60 °C with stirring. The solution is spin-coated on to a fused silica substrate at 2000 r.p.m., and dried in air for 30 min before baking on a hotplate at 100 °C for 1 h. Absorption measurements were carried out using a Cary Varian 300 spectrometer. PL emission and excitation spectroscopy of the coating were performed on a FLS980 spectrometer (Edinburgh Instruments, UK) and ultrafast luminescence measurements were performed using up-conversion spectroscopy. CL was carried out on an inclined sample in a low-vacuum field emission SEM using a 5 keV, 1.5 nA electron beam. The luminescence was collected using a reflecting objective, dispersed with a 1/8-m spectrograph and detected with an electron-multiplying CCD. X-ray photoemission spectra and Raman spectra on the coating were carried out on Kratos Axis Ultra DLD photoelectron spectrometer and Lab Ram 300 (Horiba Jobin Yvon SAS), respectively,

and more detail and analysis were provided in Supplementary Information. SEM images were taken on a Jeol 6700 F microscope. Powder sample of MABI was prepared by scratching and grinding a large area of coating on a quartz substrate. Phase purity was confirmed by powder XRD on a Panalytical expert monochromated X-ray diffractometer with Cu-$K_\alpha$ target ($\lambda = 0.15418$ nm), and unit cell parameters of the powdered sample were analysed using General Structure Analysis System software. The absorption of the powdered sample was recorded with a V650 spectrophotometer (Jasco). PL of the powder sample was measured on a Horiba Jobin Yvon spectrofluorometer and the powder was sandwiched between two quartz plates with an illuminated spot size of about 0.6 cm$^2$.

**Data availability**. All new data associated with this manuscript are available from the University of St Andrews repository http://dx.doi.org/10.17630/b8144269-6d3d-4a2a-afe2-e68fc51f962e.

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

## Acknowledgements

We thank the Engineering and Physical Sciences Research Council (EPSRC) for funding (EP/K036769/1; EP/K022237/1; EP/M024938/1). C.N. also thanks the support from Fundamental Research Funds for the Central Universities (XDJK2017B033) and Research Funding of Southwest University (20710945). We also thank Dr D. Timmerman for the kind discussion.

## Author contributions

C.N. prepared the samples and was involved in all aspects of characterisation and led the analysis and writing of the manuscript. G.H. and L.K.J. performed PL, PLE, time-resolved PL of coating; J.P. performed $CH_3NH_3I$ synthesis, single crystal and structure analysis; V.S. and C.M. performed PL of powder; L.K.J., P.E., R.M. performed CL; C.M., G.J. and D.C. performed XPS and Raman; P.M. and I.S. advised on interpretation, and D.M. and J.I. oversaw the research and worked with C.N. in elaborating the text.

## Additional information

**Competing interests:** The authors declare no competing financial interests.

