## [Peer Review File · Nature Communications]

Reviewers' comments:

Reviewer #1 (Remarks to the Author):

The manuscript is overall well written and contains important elements of novelty, provides strong experimental evidences and can be of interest to scientists in the specific field but not only. In particular, authors demonstrated for the first time the quantum cutting in MAIB that in part explain why, through the presence of localized exciton, it has poor photovoltaic properties. However, the manuscript must be modified in order to better clarify some important points of the work. It will be suitable for the publication in Nature Communications only after the requested revise.

1-The author claims there is a quantum cutting effect in the material and the PL measurements demonstrate it clearly. However, quantum cutting was demonstrated for rare-earth doped inorganic host materials where electronic levels of rare earth elements allow such a kind of phenomenon. The author should describe more in detail the electronic levels of the (Bi₂I₉)₃-clusters involved in the process, including their spectroscopic symbols.

2-When the author talks about confined clusters, it leads the reader to think that the material is showing multiple electron generation but actually is showing quantum cutting. Please clear-off this ambiguity perhaps highlighting the similarities and differences, in the specific case, of the two quantum phenomena.

Other comments are evidenced in yellow in the annotated manuscript in attachment. It includes comments to which the author must reply and modify the text accordingly.

Reviewer #2 (Remarks to the Author):

The authors have presented a detailed spectroscopic study of methylammonium bismuth iodide. Their research goals and hypotheses were certainly interesting and unique. However, there are a number of strong statements made, which I do not believe are adequately supported by the data or existing literature. I believe that this project needs to be redesigned and the results approached with less bias. The following points should be considered:

- Throughout the manuscript, the authors claim that excitons exist in methylammonium bismuth iodide. However, they do not state the exciton binding energy. As is known in the MAPbI₃ literature, it was originally thought that excitons existed before the exciton binding energy was properly measured and found to be only a few meV at room temperature (DOI: 10.1038/nphys3357). Measuring the exciton binding energy accurately is not easy, but much of the interpretation of data in the authors' manuscript depends on whether carriers exist as free electron-hole pairs or excitons in methylammonium bismuth iodide.

- There is a great deal of analogies made between the isolated Bi₂I₉³⁻ and MA⁺ groups and quantum dots. Yet, the authors provide no evidence that electrons and holes are confined in the same way. For example, the authors would need the E vs. k diagram for methylammonium bismuth iodide to prove that carriers are locally confined. What is the effective mass of electrons and holes at different points in the Brillouin zone?

- The authors create a Tauc plot in the SI where the exponent for the vertical axis is 2, implying a direct allowed transition. Yet the literature suggests that the bandgap of methylammonium bismuth iodide is indirect (DOI: 10.1002/chem.201505055), and the Tauc plot should then have an exponent of 0.5.

- The authors claim from the SEM image in Figure 2b that their thin films have large grains. But there is no proof that the microstructure they observe are single grains. Literature on MAPbI₃ and

methylammonium bismuth iodide shows that large grains in SEM can actually be composed of small crystallites, which can be observed in TEM

- The authors claim that the absorption feature at 2.44 eV is an excitonic peak. The authors need to prove that this is an excitonic peak and not due to absorption due to the split density of states in methylammonium bismuth iodide that has been reported in the literature

- In the cathodoluminescence measurements, can the authors prove that their sample was not damaged by the high energy beam? I.e.: is the lack of anisotropy in the CL measurements due to a lack of control over the incident angle as the authors claim, or because their sample changed during the measurements?

- The PL spectral features in Figure 3 are mostly sub-bandgap. Can the authors explain whether these are in fact due to defect states that emit in an anisotropic manner or are arranged in an anisotropic manner?

- It should be noted that methylammonium bismuth iodide is a very weak emitter. The authors should prove that all of the PL peaks they measured are from the material itself and not artifacts from the instrument, substrate or spectral filters

- The authors state that they dried methylammonium bismuth iodide in air. Can they prove that they do not have surface phase impurities due to reaction with air?

Finally, would it not be a cleaner experiment to perform these measurements on single crystals? Would the anisotropic measurements be less ambiguous than performing the measurements on a thin film that could have some grains not perfectly aligned?

Reviewer #1 (Remarks to the Author):

The manuscript is overall well written and contains important elements of novelty, provides strong experimental evidences and can be of interest to scientists in the specific field but not only. In particular, authors demonstrated for the first time the quantum cutting in MAIB that in part explain why, through the presence of localized exciton, it has poor photovoltaic properties. However, the manuscript must be modified in order to better clarify some important points of the work. It will be suitable for the publication in Nature Communications only after the requested revise.

1-The author claims there is a quantum cutting effect in the material and the PL measurements demonstrate it clearly. However, quantum cutting was demonstrated for rare-earth doped inorganic host materials where electronic levels of rare earth elements allow such a kind of phenomenon. The author should describe more in detail the electronic levels of the $(\text{Bi}_2\text{I}_9)^{3-}$ clusters involved in the process, including their spectroscopic symbols.

Reply: It is very true that quantum cutting has been discovered in rare-earth doped inorganic host materials (10.1126/science.283.5402.663). The quantum cutting can be achieved via the sequential emission on a single rare-earth ion or by two-step emission involving the energy transfer between the highly excited ion and another unexcited ion. What we observe in MABI is the second one, and is noted as space-separated quantum cutting since MABI contains an array of nano-sized clusters. We should also point out the charge difference between that on $\text{Bi}_2\text{I}_9^{3-}$ and that on rare-earth cations and that, in MABI, the lowest exciton state arises from excitations between the valence band, which consists of a mixture of Bi(6s) and I(5p) states, and the conduction band, which derives primarily from Bi(6p) states, and is confined zero-dimensionally in the biocuboctahedra $\text{Bi}_2\text{I}_9^{3-}$ (j.jlumin.2015.01.010). The emission from rare-earth element is mainly from the transition between 4f and 5s electrons.

2-When the author talks about confined clusters, it leads the reader to think that the material is showing multiple electron generation but actually is showing quantum cutting. Please clear-off this ambiguity perhaps highlighting the similarities and differences, in the specific case, of the two quantum phenomena.

Reply: Thanks for the comment and for bringing this to our attention so that we had the opportunity to clarify these aspects. It is true that multiple exciton generation is frequently reported for quantum dots involving the sequential emissions of a highly excited dot, but in MABI the confined clusters were arranged in a superlattice, where the electronic coupling between the closely spaced clusters will generate a phenomenon described as space separated quantum cutting[12]. Space separated quantum cutting has been demonstrated to occur between nanoclusters without rare-earth element doping and it was described in terms of a carrier multiplication phenomenon. Therefore, as per our reference [12], it is appropriate to discuss this in terms of carrier multiplication. Of course, in our case MABI presents some other advantageous characteristics, as the $\text{Bi}_2\text{I}_9^{3-}$ clusters are highly regular in size and in their spatial arrangement and orientation, which are the main reasons for angle-dependent emission and no variations in exciton binding energies among different clusters. These features are very difficult to control for nanoclusters (e.g. ref.[12]) and also for quantum cutting with rare-earth element doping. We have clarified these points in page 1."

Other comments are evidenced in yellow in the annotated manuscript in attachment. It includes

comments to which the author must reply and modify the text accordingly.

3-In the abstract, line 15-18, It must be specified that 20% of efficiency was obtained with lead halid perovskite

Reply: this has been corrected and the sentence was revised.

4-In page 2 line 58, "bioctahedron" should be "bioctahedra"

Reply: revised

5-In page 2 line 66 "methods" should be revised to "phenonmenon"

Reply: revised to "phenomena"

6-page 3 line 74-75 "This could be alleviated by surface passivation with ligands, equivalent to the organic cation in zero-dimensional hybrid compounds" The second half of this sentence is not straightforward that is referred to the organohalide hybrid compound, it should be better written.

Reply: revised. It has been suggested that this could be alleviated by surface passivation with ligands¹⁷, alternatively this also may be achieved by protecting with the organic cation in zero-dimensional hybrid organic-inorganic halides.

7-page 3 line 88, and cite nano-research 2016 9 692-702

Reply: revised and cited the reference above

8-line 103 "low-dimensional material" is redundant

Reply: deleted

9-page 5 line 134 the sentence "The stable structure of MABl could be beneficial in processing". The stable structure is not only beneficial in processing, mainly in the stability and lifetime of the device.

Reply: This sentence is revised to show the benefit of stable structure is mainly to the stability and lifetime of the device.

10-The SEM micrograph is not very exhaustive about the morphology of the film. I should be integrated with an optical microscope image of the film:

Reply: We have included optical microscope image as requested see here below (figure 2 in the manuscript):

Figure 2. (a) X-ray diffraction results from the coating and powder samples. (b) Optical microscope and (c) scanning electron micrograph of the MABl coating processed by depositing and drying of the solution of BiI_3 and $\text{CH}_3\text{NH}_3\text{I}$ in DMF. The scale bars represent $25 \mu\text{m}$.

In the optical microscope image, we can easily see the crystallite on the size of $20 \mu\text{m}$ and lying

along the substrate.

11-In other recent publications (mainly 22) the formation of the film by spin-coating resulted sem-amorphous although retain a certain crystal orientation. Can the author evidence visually the oriented crystallites? An optical image of the film can help at this purpose:

Reply: We have included an optical microscope image as requested (see reply to previous comment and figure 2) with large grains clearly visible and with darker grain boundaries. This shows the orientation of the coating, though we cannot detect the orientation by naked eyes. However, we believe that the orientation of the large crystals is confirmed and demonstrated by our XRD measurements which contain sharp peaks corresponding to the given directions, similarly to the analysis in ref 22 There could be areas of different orientation, but the effect is minor, as they would otherwise result in the XRD pattern.

12-page 6 line 154-158. It would be useful to remark the difference with MAPbI₃ and similar where the absorption edge is affected by the cation: MA, FA, Cs..

Reply: The substitution of Cs or FA on MA site in MAPbI₃ has been shown to cause significant shift (>100 meV) of the band edge (10.1039/c3ee43822h, 10.1002/adma.201501978), however this is different from the MABl system where nearly no band edge shift was found if MA is replaced by C₆H₁₁NH₃⁺ (10.1016/j.jlumin.2015.01.010) and similarly for Cs (10.1002/adma.201501978). We have included a remark about this point as suggested at page 6. 13-page6 line 167, the definition quantum wells is not correct, they are rather quantum dots

Reply: revised to "quantum dots".

14-page6 line168, "or a core-shell structure" better to write resembling or miming a core-shell structure

Reply: revised.

15-Page 6 line 171"distributed randomly" in the perpendicular direction. Actually they do not distribute randomly, but according to lattice parameters.

Reply: What we trying to convey by saying "distributed randomly" is focused on the angle of the crystals, so we revise it to "orientated randomly".

16-page 6 line 154-156, the authors should briefly explain the reason why to compare with (C₆H₁₁NH₃)₃Bi₂I₉

Reply: we compare MABl with (C₆H₁₁NH₃)₃Bi₂I₉ because they show iso-structure with only difference in organic cations.

17-page 8-page line line 232-234. This sentence must be referenced. For instance, in ref. 22 the MAIB crystal was already described as media hosting Bi₂I₉- clusters.

Reply: This sentence is referred to 22

18-page 8 line 230 "anisotropic" should be revised to "optical anisotropy"

Reply: revised to " optical anisotropy "

19-Page 8 line 197-199. The scheme in Figure 4 should be improved. E.g. insert in the 3D crystal lattice drawing planes as separators.

Reply: This has been improved according to the reviewer suggestions.

20-page 9 line 248-249. Is this PL of the powder? must be specified in the captions.

Reply: revised.

Reviewer #2 (Remarks to the Author):

The authors have presented a detailed spectroscopic study of methylammonium bismuth iodide. Their research goals and hypotheses were certainly interesting and unique. However, there are a number of strong statements made, which I do not believe are adequately supported by the data or existing literature. I believe that this project needs to be redesigned and the results approached with less bias. The following points should be considered:

Reply: We have tried to address this comment by clarifying our arguments and conclusions in both the response and the article and have attempted to moderate some of the statements to make sure that we retain the appropriate balance. With the promising energy conversion efficiency from MAPI, the development of lead-free perovskite or non-perovskite structure hybrid materials excites great interest in the community. The recent results on MABI as absorber showed low efficiency but with better stability than MAPI, and the relation between the structure and power output could be understood by the behavior of excitons. The existence of $\text{Bi}_2\text{I}_9^{3-}$ structure is based on the crystal structure without any bias and the localization of exciton is proven by the binding energy shown in this study and previous literature. We approached to the conclusion based on extensive spectroscopy measurements and detailed reading and citation of literature as analyzed below.

-Throughout the manuscript, the authors claim that excitons exist in methylammonium bismuth iodide. However, they do not state the exciton binding energy. As is known in the MAPbI₃ literature, it was originally thought that excitons existed before the exciton binding energy was properly measured and found to be only a few meV at room temperature (DOI: 10.1038/nphys3357). Measuring the exciton binding energy accurately is not easy, but much of the interpretation of data in the authors' manuscript depends on whether carriers exist as free electron-hole pairs or excitons in methylammonium bismuth iodide.

Reply: The key difference between MABI and methylammonium lead halide lies in the crystal structure: one contains a 3-D framework of PbI_6 octahedra while MABI shows clusters of Bi_2I_9 and the I-M-I (M=transition metals) framework is crucially important to the transport of charge carriers. The exciton binding energy has been measured and reported in the literature to be in the region of 70-300 meV. Kawai et al. (10.1002/pssb.2221770128) studied the absorption coefficient of MABI single crystals of different thickness at different temperatures, and the exciton binding energy was calculated to be 300 meV, which is much larger than methylammonium lead halide. Park et al. (10.1002/adma.201501978) studied the solar cell performance as well as the PL and absorption and found an exciton binding energy of 70 meV, with the binding energy increased to 300 meV when Cs replaced methylammonium or with partial doping of Cl on I position. With large the binding energy as such, it would not be so subtle to be measured.

In this study, we can also determine the exciton binding energy (330 meV) since we have measured the absorption and PL maxima at different angles. Most importantly, the analysis of binding energy from the PL on coating is in agreement with the Stokes shift determined from the powdered sample, as shown in Figure 5(b).

-There is a great deal of analogies made between the isolated $\text{Bi}_2\text{I}_9^{3-}$ and MA^+ groups and quantum dots. Yet, the authors provide no evidence that electrons and holes are confined in the same way. For example, the authors would need the E vs. k diagram for methylammonium

bismuth iodide to prove that carriers are locally confined. What is the effective mass of electrons and holes at different points in the Brillouin zone?

Reply: The confinement of charge carriers because of the organic group is not unprecedented (Reference 6. [10.1126/science.aac7660](https://doi.org/10.1126/science.aac7660)) and is referred to in the submitted version, but is studied mostly in layered perovskite with organic group in between layers of PbI_6 slabs. We used experimental results and in particular, PLE to demonstrate that the holes and electrons were confined due to the variation of PLE with angle for the same emission peak and the variation of PLE for different emission wavelengths as shown in Figure 4. E vs. k diagram would be ideal for better understanding but would not change the basis of our point to make.

-The authors create a Tauc plot in the SI where the exponent for the vertical axis is 2, implying a direct allowed transition. Yet the literature suggests that the bandgap of methylammonium bismuth iodide is indirect (DOI: [10.1002/chem.201505055](https://doi.org/10.1002/chem.201505055)), and the Tauc plot should then have an exponent of 0.5.

Reply: There was some controversy on the nature of this transition in literature. The paper ([10.1002/chem.201505055](https://doi.org/10.1002/chem.201505055)) suggested an indirect band transition for this peak, but Park et al. ([10.1002/adma.201501978](https://doi.org/10.1002/adma.201501978)), Lyu et al. ([10.1007/s12274-015-0948-y](https://doi.org/10.1007/s12274-015-0948-y)) and Abulikemu et al. ([10.1039/C6TA04657F](https://doi.org/10.1039/C6TA04657F)) used the exponent of 2 for the determination of bandgap. We have therefore included reference to these publications highlighting disagreement in the literature. However, we have produced a more detailed analysis of the bandgap transition, shown in Figure S4, which provides some clarity on this aspect. Our analysis based on the experimental results and now included in the SI, suggests a direct bandgap behavior.

-The authors claim from the SEM image in Figure 2b that their thin films have large grains. But there is no proof that the microstructure they observe are single grains. Literature on MAPbI_3 and methylammonium bismuth iodide shows that large grains in SEM can actually be composed of small crystallites, which can be observed in TEM

Reply: Indeed the literature has reported a range of grain sizes, small but also large (e.g. mm-size for high performance solar cells). In our case, the high intensity and sharp XRD peaks are quite clear that the grains are indeed large and would imply that the contribution of anisotropy from quantum confinement of small grains should be minor.

-The authors claim that the absorption feature at 2.44 eV is an excitonic peak. The authors need to prove that this is an excitonic peak and not due to absorption due to the split density of states in methylammonium bismuth iodide that has been reported in the literature

Reply: There is controversy in literature on the nature of this peak : it is ascribed to the split conduction band density of states after DFT calculations ([10.1002/chem.201505055](https://doi.org/10.1002/chem.201505055)), while there were also literature on the excitonic nature of the absorption peak, e.g. [j.lumin.2015.01.010](https://doi.org/10.1021/jlumin.2015.01.010) with vibration experiment and DFT calculations and [10.1002/pssb.2221770128](https://doi.org/10.1002/pssb.2221770128) with absorption coefficients of single crystals at different temperatures. We believe that this peak is an excitonic peak as it shows direct band behavior while the binding energy of excitons is high.

-In the cathodoluminescence measurements, can the authors prove that their sample was not damaged by the high energy beam? I.e.: is the lack of anisotropy in the CL measurements due to a lack of control over the incident angle as the authors claim, or because their sample changed during the measurements?

Reply: CL has been performed on methylammonium lead halide ([acs.jpcc.5b09698](https://doi.org/10.1021/acs.jpcc.5b09698)), and

there is damage of beam on the emission of the perovskite at high-energy beam, but we do think it is a minor issue for our sample, since the two runs done at different angles on the same sample showed no extra peaks and are quite similar to each other. The argument of degradation for MAPI in the literature was based on the assumption that the first few runs were reliable, as the claim of degradation was somewhat weak if it had started at the beginning. Moreover, we used 1.5 nA, current which is much smaller than the cited literature which used 12 nA for the degradation experiment. The electron beam at 5 kV would not change the angle of sample because the sample is stable enough to allow a clear SEM image, as shown in Figure 2(c).

-The PL spectral features in Figure 3 are mostly sub-bandgap. Can the authors explain whether these are in fact due to defect states that emit in an anisotropic manner or are arranged in an anisotropic manner?

Reply: In order to have an angle-dependent emission from defects, they should arrange in an anisotropic manner because if defects emitting anisotropically or isotropically were not regularly arranged, the final emission will be a collective emission from different directions, and will not vary significantly from angle to angle. The PL anisotropy in our case is different from PL polarization.

-It should be noted that methylammonium bismuth iodide is a very weak emitter. The authors should prove that all of the PL peaks they measured are from the material itself and not artifacts from the instrument, substrate or spectral filters

Reply: The reviewer is correct that the emission from MABI is not exceptional, but the emission of MABI has been observed and reported in recent literature (10.1002/chem.201505055; 10.1002/adma.201501978; 10.1039/C6TA04657F). We used different instruments, one at St Andrews and the other at AIST, and they corroborating each other since the PL of coating powder, and the Stokes shift in the PL of powdered sample can be derived from the peak maxima in that of coating, as shown in Figure 5(b).

-The authors state that they dried methylammonium bismuth iodide in air. Can they prove that they do not have surface phase impurities due to reaction with air?

Reply: We proved that the bulk materials were in the right composition as there is no extra peak in the XRD and the surface is analyzed with XPS and Raman spectroscopy in current version, as shown in Figure S2, Figure S3 and Table S1.

Finally, would it not be a cleaner experiment to perform these measurements on single crystals? Would the anisotropic measurements be less ambiguous than performing the measurements on a thin film that could have some grains not perfectly aligned?

Reply: Of course, there are some grains not perfectly aligned as shown even in our SEM, but the XRD shows that their effect is minor. We actually synthesized single crystal, but the thin sheet of crystal is fragile to handle and not perfectly flat or big enough to show strong PL signals. A single thicker crystal would present similar issues, if not worse, in the misalignment during the PL.

Reviewers' comments:

Reviewer #1 (Remarks to the Author):

The author replied exhaustively to all the points and corrected the issues in the manuscript, however one point still remains unsolved. In the sentence from line 172 to line 176 authors that Bi2I9 dimers are distributed randomly in the crystal in the direction perpendicular to the main axis, that is not true. Nevertheless, in the reply to reviewers authors (point 16) declares that the crystals are distributed randomly, that is correct. Authors must modify the text in coherence with what is reported in the "reply to reviewers", namely that crystals are distributed randomly and not Bi2I9 dimers.

Reviewer #2 (Remarks to the Author):

The authors have put together a very detailed response to the comments and have made many good revisions. The work is improved, but the authors still need to address the following comments:

1. The work by Kawai et al. is reasonably convincing in suggesting methylammonium bismuth iodide to be excitonic, particularly the temperature-dependent data in Fig. 2. However, the authors do not talk about this until page 6. In my opinion, the paper is reliant on this data and should be mentioned in the first paragraph of the introduction. Also, an alternative explanation for the observed absorption spectra by Kawai et al. is that the conduction band density of states is split or partially split. This can be found in the DOS calculations in 10.1007/s12274-015-0948-y and 10.1002/chem.201505055. The authors should discuss in the main text (in the results and discussion section) whether the result by Kawai et al. is more consistent with excitonic behavior or the CB DOS calculated independently by two groups, and how confident they are, based on their data and analysis. I believe it is important that this is in the main text so that other groups are aware of these two alternative hypotheses and can make up their own mind over whether they believe the authors' argument.

2. The authors' argument of confined charge-carriers could be improved by analyzing the E vs. k diagram. This has been provided in Fig. 4c of 10.1007/s12274-015-0948-y, which shows flat bands in some regions (indicating high effective mass and no transport) and disperse bands in others (indicating there is some transport). Can the authors map out the Brillouin zone of methylammonium bismuth iodide and confirm that the flat regions are between the Bi2I93-clusters?

3. The authors argue that the bandgap of MABI is direct. Going through their listed literature:
a. 10.1002/chem.201505055 claims the bandgap to be indirect based the calculated zero dipole matrix element. But they show that the direct transition is 0.1 eV from the indirect bandgap
b. 10.1002/adma.201501978 does not seem to justify why they chose a direct bandgap for their materials. They say "the optical bandgaps are estimated to be around ..." This gives me the impression that they guessed the bandgaps to be direct
c. 10.1007/s12274-015-0948-y actually claims MABI to have an indirect bandgap. This was based on the E vs. k diagram, which seems to show the VBM to be between the gamma and F points, and CBM to be at the gamma point. But given how flat the VB is, it isn't too surprising that the direct transition is close to the indirect transition
d. 10.1039/C6TA04657F states the different coefficients in a Tauc plot for direct and indirect bandgaps, but does not seem to justify why they chose to use a direct bandgap. This gives me the impression that they also made a guess

From the above, it would seem that the discrepancy in the literature over whether the bandgap is direct or indirect is split between people who calculated the band structure and those who guessed. I am also not convinced by the authors' analysis in Fig. S4d – how can they rule out that

these states are not defects, especially since they made the films by solution processing at low homologous temperatures? The authors need to re-consider their analysis and the implications of an indirect bandgap.

4. Can the authors quantify their grain size from XRD? Can they obtain the instrument broadening, then determine the grain size? This may be difficult to do over the whole pattern since many of the peaks measured are overlaps of multiple peaks, but perhaps the authors could either perform Scherrer analysis on a few measured peaks that only comprise of one fitted peak to estimate the grain size, or fit the XRD pattern with the CIF and use the FWHM of the fit to perform Williamson-Hall analysis.

5. It is appreciated that the authors measured their sample in two labs. But to ensure that the measured PL is not due to the substrate, could the authors measure the substrate they used with the same instruments and the same settings? These measurements of the substrate should be performed at the different angles.

6. Thank you to the authors for adding XPS data in Fig. S2. In their rebuttal letter they state that this proved their material to have the right composition, but Fig. S2 clearly shows there to be a distinct O 1s peak. What is the origin of this peak? Does this mean that their surface is not purely $(\text{CH}_3\text{NH}_3)_3\text{Bi}_2\text{I}_9$? Would this influence the PL measurements, e.g.: by introducing extra peaks? What are the possible surface phase impurities and at what wavelength do those materials emit?

7. In Fig. 5d, the authors should quantify the PLOQY.

Response to Reviewers' comments:

Reviewer #1 (Remarks to the Author):

The author replied exhaustively to all the points and corrected the issues in the manuscript, however one point still remains unsolved. In the sentence from line 172 to line 176 authors that Bi₂I₉ dimers are distributed randomly in the crystal in the direction perpendicular to the main axis, that is not true. Nevertheless, in the reply to reviewers authors (point 16) declares that the crystals are distributed randomly, that is correct. Authors must modify the text in coherence with what is reported in the "reply to reviewers", namely that crystals are distributed randomly and not Bi₂I₉ dimers.

Reply: We appreciate the reviewer pointing this out. To clarify our meaning we replace the "Bi₂I₉³⁻ clusters" with "MABI crystals"

Reviewer #2 (Remarks to the Author):

The authors have put together a very detailed response to the comments and have made many good revisions. The work is improved, but the authors still need to address the following comments:

1.The work by Kawai et al. is reasonably convincing in suggesting methylammonium bismuth iodide to be excitonic, particularly the temperature-dependent data in Fig. 2. However, the authors do not talk about this until page 6. In my opinion, the paper is reliant on this data and should be mentioned in the first paragraph of the introduction. Also, an alternative explanation for the observed absorption spectra by Kawai et al. is that the conduction band density of states is split or partially split. This can be found in the DOS calculations in 10.1007/s12274-015-0948-y and 10.1002/chem.201505055. The authors should discuss in the main text (in the results and discussion section) whether the result by Kawai et al. is more consistent with excitonic behavior or the CB DOS calculated independently by two groups, and how confident they are, based on their data and analysis. I believe it is important that this is in the main text so that other groups are aware of these two alternative hypotheses and can make up their own mind over whether they believe the authors' argument.

Reply: We have now cited the work of Kawai et al in the introduction at a point where we felt it was most appropriate (see page 3). Also we have expanded our discussion as suggested by the reviewer in the "Results and discussion" section at page 6 where the papers indicated by the reviewer are explicitly addressed.

2.The authors' argument of confined charge-carriers could be improved by analyzing the E vs. k diagram. This has been provided in Fig. 4c of 10.1007/s12274-015-0948-y, which shows flat bands in some regions (indicating high effective mass and no transport) and disperse bands in others (indicating there is some transport). Can the authors map out the Brillouin zone of methylammonium bismuth iodide and confirm that the flat regions are between the Bi₂I₉-clusters?

Reply: Indeed this reference does show some flat band regions which would indicate low

long range transport. It is probably a little risky to align reciprocal space band structures with real space structures. Certainly directional information is available and it does seem that the high reciprocal mass is consistent with electrons constrained in clusters without so much long range mobility.

3. The authors argue that the bandgap of MABI is direct. Going through their listed literature:

a. 10.1002/chem.201505055 claims the bandgap to be indirect based on the calculated zero dipole matrix element. But they show that the direct transition is 0.1 eV from the indirect bandgap

b. 10.1002/adma.201501978 does not seem to justify why they chose a direct bandgap for their materials. They say the optical bandgaps are estimated to be around ... This gives me the impression that they guessed the bandgaps to be direct

c. 10.1007/s12274-015-0948-y actually claims MABI to have an indirect bandgap. This was based on the E vs. k diagram, which seems to show the VBM to be between the gamma and F points, and CBM to be at the gamma point. But given how flat the VB is, it isn't too surprising that the direct transition is close to the indirect transition

d. 10.1039/C6TA04657F states the different coefficients in a Tauc plot for direct and indirect bandgaps, but does not seem to justify why they chose to use a direct bandgap. This gives me the impression that they also made a guess

From the above, it would seem that the discrepancy in the literature over whether the bandgap is direct or indirect is split between people who calculated the band structure and those who guessed. I am also not convinced by the authors' analysis in Fig. S4d.— how can they rule out that these states are not defects, especially since they made the films by solution processing at low homologous temperatures? The authors need to re-consider their analysis and the implications of an indirect bandgap.

Reply: Since the DFT calculation on MABI assumes that there is no charge localization as in normal semiconductors, we can see that the calculated results point to an indirect band gap transition. However, our results indicate that there was substantial anisotropy and localization in this material as can be seen from PL, PLE, transient PL and CL, and it is reasonable to conceive a direct band gap for the excitonic peak. As an indirect band transition semiconductor does not conflict the excitonic peak, the indirect band transition could be assigned to the continuous absorption on the blue side of the excitonic peak (Figure 1). The nature of continuous absorption cannot be validated in our study due to its overlapping with the excitonic peak, but Kawai et al provided the data at low temperature showing a discontinuous absorption, which is closer to the studied temperature (at 0 K) in DFT calculations.

Figure 1. Absorption spectra for MABI at different temperatures from the work of Kawai and et al.

Since the absorption coefficient for sample with different crystal orientation (10.1039/C6TA04657F) due to the localization of charge carriers, the omission of absorption peak is not surprising for the absorption from powder or measured with reflectance spectroscopies in 10.1007/s12274-015-0948-y and Figure 4 (a, b) in our study that provides the absorption without information on orientation.

We think the result on Figure S4d is reliable as we used XPS to prove the composition, XRD to validate structure and Raman to show the vibration and energy transfer. The crystals are quite large and robust and the stoichiometry is a simple molecular ratio, so it seems unlikely that extrinsic defects will cause any problems. More importantly, defects are very unlikely to enlarge the band gap as an indirect transition would give smaller band gap than Figure S4d presents, since they tend to be contribute to the intraband states.

4.Can the authors quantify their grain size from XRD? Can they obtain the instrument broadening, then determine the grain size? This may be difficult to do over the whole pattern since many of the peaks measured are overlaps of multiple peaks, but perhaps the authors could either perform Scherrer analysis on a few measured peaks that only comprise of one fitted peak to estimate the grain size, or fit the XRD pattern with the CIF and use the FWHM of the fit to perform Williamson-Hall analysis.

Reply: The grain sizes are observed in microscopy to be far larger than the effective region since the Scherrer function can be only applied to nanoscale particles, therefore we decided not to include this analysis in our manuscript. In fact, if we select two peaks for the calculation and fitting parameters for the peaks at 24.5° and at 29.1° the result (Table 1, and Figure 2) show that the crystals were on the scale of micrometer, which confirm large particle size. The FWHM of the instrumental response function (IRF) on standard Si substrate at 28° was subtracted for the peak widening.

Table1. Parameters for the determination of crystal size using Scherrer function

λ / nm	2θ / $^\circ$	FWHM / $^\circ$	FWHM / $^\circ$ IRF	$\Delta(2\theta)$ / $^\circ$	$\Delta(2\theta)$ /rad	size / nm
0.15	24.57901	0.06829	0.061	0.00729	0.0001272	2414.31615
0.15	29.13374	0.10127	0.061	0.04027	0.0007025	441.2225992

Figure 2. XRD of powdered sample with Gaussian fitting (red lines).

These values are micron scale as expected and are only an illustration as this analysis can only say that the particles are much larger than nanoscale.

5. It is appreciated that the authors measured their sample in two labs. But to ensure that the measured PL is not due to the substrate, could the authors measure the substrate they used with the same instruments and the same settings? These measurements of the substrate should be performed at the different angles.

Answer: Following the referee's suggestion, a fused silica substrate was remeasured on the Edinburgh Fluorimeter Instrument, FLS980 at University of St Andrews, using the original scan parameters (2 nm slit width, 2 nm step size, excitation wavelength of 350 nm, dwell time 0.5 s). These data are shown in red in below PL spectra. For comparison the previously measured spectra of the MABI samples are shown in black.

The PL responses of the substrate were similarly studied at AIST. The PL intensity of the bare substrate was very weak and no obvious peak was observed in the region of interest, ensuring that the PL is not from the substrate

6. Thank you to the authors for adding XPS data in Fig. S2. In their rebuttal letter they state that this proved their material to have the right composition, but Fig. S2 clearly shows there to be a distinct O 1s peak. What is the origin of this peak? Does this mean that their surface is not purely (CH₃NH₃)₃Bi₂I₉? Would this influence the PL measurements, e.g.: by introducing extra peaks? What are the possible surface phase impurities and at what wavelength do those materials emit?

Reply: As we stated in the supplementary information, this O1s peak, after the binding

energy, is due to the absorbed oxygen species rather than those in the lattice. The O1s peak due to adsorbed species at the surface is regularly observed in XPS analysis unless a vacuum sample transfer is used. This has not influenced our analysis and discussion as exposure to air has shown no degradation in our films, also confirmed by N-, C- and Bi-peaks. In addition, if there were emission from the absorbed species in the detecting range, there would be a constant peak in the PL at different angles if it is so strong, which we did not observe.

7. In Fig. 5d, the authors should quantify the PLQY.

Reply: We thank the reviewer for the suggestion; unfortunately one is from an integrating sphere whilst the other is from a normal light detector. Thus values can only be estimated. However, in the context of this manuscript, however, we do not feel that the PLQY is required. Our discussion and conclusions are supported by the experimental evidence provided and do not rely on the quantification of the PLQY.

Reviewers' comments:

Reviewer #2 (Remarks to the Author):

The authors have now addressed almost all of my previous comments. There is, however, one minor point that need to be addressed prior to publication. The authors claim that the O 1s peak they observe in XPS is due to adsorbed surface oxygen. They need to show depth-resolved XPS and mill at least 50 nm to show that there is no O in the bulk. They also should quantify the at.% of each element.

Reviewer #2 (Remarks to the Author):

The authors have now addressed almost all of my previous comments. There is, however, one minor point that needs to be addressed prior to publication. The authors claim that the O 1s peak they observe in XPS is due to adsorbed surface oxygen. They need to show depth-resolved XPS and mill at least 50 nm to show that there is no O in the bulk. They also should quantify the at.% of each element.

The XPS peak at 531.85 eV is not generally consistent with a metal oxide as the value is too large, so it is not the most likely degradation product, i.e. a bismuth oxide species. Hoyer et al (10.1002/chem.201505055) clearly demonstrated an extra oxygen peak will be found at a lower binding energy (than the one we report here), when they looked at a sample aged in ambient air, indicating that oxygen is indeed included as metal oxide. The peak we observe at 531.85 eV could relate to a carbonate or hydroxy species and is a good fit for hydroxyl terminated Molybdenum, so this oxygen peak may be a stray peak from the substrate although adsorbed water is the more likely explanation. Furthermore in all the recent references with XPS data on MABI (10.1002/adma.201501978, 10.1007/s12274-015-0948-y and 10.1002/chem.201505055), a similar O1s peak can be clearly distinguished.

More conclusive than XPS are the unit cell parameters measured by X-ray powder diffraction which are exactly as expected from the known crystal structure (see also refined pattern in separate file 531.85 eV), which is very strong confirmation that the bulk structure consists of $(\text{CH}_3\text{NH}_3)_3\text{Bi}_2\text{I}_9$. Hence, we do not think there is need to be concerned that there is oxygen in the MABI lattice for the samples reported in this study.

As requested We checked the XPS content of N/Bi/I from the XPS and the atom ratio is 3.03/2/4.45, which is close to the theoretical ratio and this is now reported in the supplementary.

Reviewers' comments:

Reviewer #2 (Remarks to the Author):

I agree with the authors' rebuttal on the O1s peak position. I have checked against the Handbook for X-ray photoelectron spectroscopy by John Moulder and agree that the binding energy would correspond to adsorbed surface species. The current text in the supporting information is adequate.

I am, however, confused by the authors' report of the stoichiometric ratio. The N:Bi:I ratio should be 3:2:9, as the authors state [(CH₃NH₃)₃Bi₂I₉]. Yet the authors report 3:2:4.5

Is this an error or is this actually what the stoichiometric ratio is? The authors need to clarify.

The referee's comment was I agree with the authors' rebuttal on the O1s peak position. I have checked against the Handbook for X-ray photoelectron spectroscopy by John Moulder and agree that the binding energy would correspond to adsorbed surface species. The current text in the supporting information is adequate.

I am, however, confused by the authors' report of the stoichiometric ratio. The N:Bi:I ratio should be 3:2:9, as the authors state [(CH₃NH₃)₃Bi₂I₉]. Yet the authors report 3:2:4.5

Is this an error or is this actually what the stoichiometric ratio is? The authors need to clarify.

Our response is

The iodine content should be doubled to 8.90. This has been rectified in the supporting material and is highlighted in yellow. Many thanks for spotting this oversight/mistake,